# Randomised controlled trial to investigate the effectiveness of thoracic epidural and paravertebral blockade in reducing chronic post-thoracotomy pain (TOPIC): a pilot study to assess feasibility of a large multicentre trial

Joyce Yeung,[1,2] Lee Middleton,[3] Kostas Tryposkiadis,[4] Amy Kerr,[5] Jane Daniels,[6] Babu Naidu,[5,7] Teresa Melody,[2] Andreas Goebel,[8] Matthew Wilson,[9] Sajith Kumar,[2] Lajos Szentgyorgyi,[10] Sarah Flanagan,[11] Rajesh Shah,[12] Antony Worrall,[2] Fang Gao[2,7]

For numbered affiliations see end of article.

**Correspondence to**
Professor Fang Gao;
f.gaosmith@bham.ac.uk

## ABSTRACT

**Objectives** Thoracotomy is considered one of the most painful surgical procedures. The incidence of chronic post-thoracotomy pain (CPTP) is up to 50%. Paravertebral blockade (PVB) may be superior to thoracic epidural blockade (TEB) in preventing CPTP. The specific objective of this pilot study was to assess the feasibility of conducting a larger trial to determine whether PVB at thoracotomy is more effective in reducing CPTP compared with TEB.

**Design** A randomised, parallel, external pilot study was conducted to assess whether a large randomised trial of TEB and PVB with CPTP as the primary outcome is feasible.

**Setting** Two adult thoracic centres in the UK.

**Participants** All adult patients admitted for elective open thoracotomy. Participants were excluded if they were American Society of Anesthesiologists physical status IV or V; or if there is contraindication to local anaesthetics; infection near the proposed puncture site; coagulation/thoracic spine disorders; required chest wall resection or emergency thoracic surgery or had a previous thoracotomy.

**Results** All patients presenting for thoracotomy were screened over a 12-month period with 194 found to be eligible. Of these, 69 (36%) were randomised (95% CI 29% to 42%). Discounting five participants who died, 54 of 64 participants (84%) returned questionnaire booklets at 6 months. The number of participants indicating at least a moderate level of chest pain at 6 months was lower with PVB but with high levels of uncertainty (RR: 0.7; 95% CI 0.3 to 1.7 for worst pain; RR: 0.3; 95% CI 0.0 to 2.8 for average pain). There were no safety concerns.

**Conclusions** A large, multicentre randomised controlled trial of PVB versus TEB is feasible as it is possible to randomise and follow up participants with high fidelity. Pain scores were lower on average with PVB compared with TEB but a much larger trial is required to confirm this reliably.

### Strengths and limitations of this study

► This randomised, external, pilot trial had preplanned feasibility thresholds to assess whether a large trial is feasible.

► Selected two sites represented the national thoracic and anaesthetic cohort, providing best representation of full trial national recruitment.

► Consultant anaesthetists in both participating sites had adequate training and appropriate assessment for the competence of each technique.

► The study was not designed and powered to be large or long enough to provide convincing evidence to support the use of either thoracic epidural blockade or paravertebral blockade as intraoperative anaesthetic technique, but provides evidence that a larger, substantive trial is feasible.

**Trial registration number** ISRCTN45041624

## BACKGROUND

An estimated 7200 thoracotomies (surgical incision into the chest wall) are performed annually in the UK, most commonly to treat lung cancer.[1] Thoracotomy is considered one of the most painful surgical procedures due to tissue, muscle and nerve damage from the incision and wound retraction.

Intercostal nerve injury can result in a high risk of persistent pain for months after surgery.[2–4] The incidence of chronic post-thoracotomy pain (CPTP; defined as pain that recurs or persists at least 2 months following the surgery)[5] is thought to be as high as 50%.[6]

Two main analgesic techniques are commonly used for perioperative pain control during thoracotomy. Thoracic epidural blockade (TEB) blocks nerves that supply the chest with local anaesthetic bilaterally, at spinal cord level. It acts by reducing onward transmission of painful nerve signals but may not abolish them completely.[7 8] Paravertebral blockade (PVB) involves injecting local anaesthetic into the paravertebral space, which contains spinal nerves (and sometimes even extension of the dura), white and grey rami communicantes, the sympathetic chain and intercostal vessels, on the side of surgery.[9] There are studies that describe the spread of PVB injections anteriorly across the heads and necks of the ribs to the spaces above and below; medially through an intervertebral foramen or spread laterally in the intercostal plane.[9] However, compared with TEB, PVB has the potential to completely block painful nerve signals from reaching the spinal cord.[9 10] This total blockade of nerve signals could remove the stimulus for 'central sensitisation', which underpins the formation of chronic pain pathways. PVB could be uniquely effective in preventing long-term pain,[11] and there is evidence from a recent trial of two techniques in breast surgery to support this premise.[12] With limited current evidence to support the most effective choice of anaesthetic technique in preventing CPTP, current UK practice varies greatly.[13] A recent Cochrane review recommended that a high-quality randomised controlled trial (RCT) to compare TEB and PVB with the primary outcome of chronic pain is urgently needed.[14]

The overall aim of this research is to determine whether PVB at thoracotomy is more effective in reducing CPTP compared with TEB. To answer this question, a large, multicentre RCT is required. The specific aim of this pilot study was to assess the feasibility of conducting a larger trial and to enhance the likelihood of its success by developing the necessary structure and processes that a large trial would need. Objectives included an assessment of the effectiveness of the patient identification and screening process; identification of reasons for failure to randomise; development of educational and training materials for surgeons and anaesthetists; evaluation of robustness of in-hospital data collection processes; assessment of trial processes including impact on participants and staff.

## METHODS
We conducted a randomised, parallel-group, external pilot trial of TEB versus PVB for perioperative pain control during thoracotomy. Recruitment took place over 12 months in two adult thoracic centres: Heartlands Hospital, Birmingham and University Hospital South Manchester (Wythenshawe, England).

### Population and inclusion/exclusion criteria
Eligible participants were all adults admitted for an elective open thoracotomy. Participants were excluded if they had any of the following: American Society of Anesthesiologists (classification of fitness) physical status IV or V (as these patients are unlikely to present for elective surgery due to their physical condition), contraindication to local anaesthetics, infection near the proposed puncture site, coagulation abnormalities (in accordance with Association of Anaesthetists in Great Britain and Ireland guidance 2013), thoracic spine disorders, required chest wall resection or emergency thoracic surgery or had a previous thoracotomy (scarring due to previous surgery can limit the effectiveness of paravertebral block and these patients may have existing chronic pain).

### Study conduct
All patients thought to fulfil the eligibility criteria were approached with study information at their preoperative assessment clinic. They were provided with a Patient Information Sheet and given the opportunity to consider participation. Once eligibility was confirmed and written informed consent was obtained, randomisation was performed prior to surgery using a web-based central randomisation system (via Birmingham Clinical Trials Unit) to allocate patients to either TEB or PVB in a 1:1 ratio. Minimisation was used to achieve balance between sex, age (<65 years or ≥65 years), thoracotomy for lung cancer resection or for other indication and study site. Research nurses at each site were responsible for randomisation and assigned group allocation was revealed to responsible anaesthetist when patients arrived at the operating theatre. An Oversight Committee was formed to provide independent guidance to the Trial Management Committee and to review accruing safety information during the period of recruitment. Public and patient involvement was integral to the study throughout.

### Interventions
Interventions were delivered by experienced thoracic anaesthetists, trained and deemed competent in both anaesthetic techniques. Participants were not told specifically about their allocated intervention. It was not possible to blind anaesthetists, surgeons or nursing staff due to the nature of the interventions. PVB was performed using three single injections preincision with the patient awake or asleep, at the levels of T3–4, 5–6 and 7–8 with 15 mL 0.25% levobupivacaine/bupivacaine, using a landmark technique at each level. The PVB catheter was then placed at T5 under direct vision by a surgeon during surgery. A loading dose of 10 mL 0.25% levobupivacaine/bupivacaine was administered before chest closure, followed by infusion of 0.125% levobupivacaine/bupivacaine 0.1–0.25 mL/kg/hour until end of operation. TEB was inserted preincision with the patient awake or asleep, with a catheter inserted at the spinal level supplying the skin at the incision site (normally T5–6), a test dose of 3 mL of 0.5% bupivacaine, and a loading dose of 0.25% levobupivacaine/bupivacaine 0.1 mL/kg with up to 3 mg of diamorphine. This was followed by infusion of 0.125% levobupivacaine/bupivacaine with 2 µg/mL fentanyl at 0.1–0.25 mL/kg/hour. Further information on delivery

and training on the interventions is available in the study protocol.[15] Identical dressings and pumps were used for both interventions to maintain group allocation.

## Pilot outcomes

The following outcomes and targets were set a priori as being indicative that a larger trial would be feasible to conduct.[15] These were as follows: (1) patient recruitment rate (the proportion of eligible patients randomised) at least 25%; (2) screening rate (the proportion of all patients listed for elective thoracotomy that were screened for eligibility and recorded on a screening log) at least 90%; (3) clinicians' willingness to recruit: national survey of consultant thoracic anaesthetist indicating the number willing to participate in a larger study at least 70%; (4) data completion—number of anaesthetic/perioperative forms completed at least 90% and follow-up questionnaires completed at 6 months at least 80%. The study was given a 12-month recruitment period, after which screening would stop.

## Clinical and participant-reported outcome measures

Baseline data (including participant completed questions as indicated below) were recorded at the preoperative assessment appointment. Intraoperative (anaesthetic technique and use), postoperative and discharge data (analgesic use, pain scores on days 1–3, acute complications, mortality and length of stay) were collected by the study team. Three-month and 6-month post-randomisation follow-up was conducted by postal questionnaire. This included Visual Analogue Scales (VAS)[16] for worst and average chest pain (overall, at rest, after coughing, after moving, after physiotherapy; with higher score—maximum 10—indicating higher levels of pain, as marked along a 10 cm line), Brief Pain Inventory (BPI),[17] Neuropathic Pain Scale (NPS),[18] generic health-related quality of life (EQ-5D-5L),[19] Hospital Anxiety and Depression Scale[20] and assessment of satisfaction and participant blinding. Serious adverse events were collected throughout. Patient outcome data were collected by a research team member who was blinded to the assigned group.

## Statistics

We expected to recruit between 50 and 75 participants over a 12-month recruitment period, depending on the number found to be eligible. We estimated that there would be approximately 500 open elective thoracotomies from the two sites, of which 60% would potentially be eligible (300). Using our own target criteria of 25% recruited would yield 75 participants. This number would be sufficient to measure the recruitment rate to uncertainty width up to approximately 10%.

Feasibility outcomes were considered with simple summary statistics (proportions and percentages), with uncertainty estimates provided by 95% CIs. Clinical and participant-reported outcome measures were analysed with point estimates (RR or mean difference) and 95%

CIs, adjusting for the minimisation variables. They were not subject to hypothesis testing as the size of the sample would not allow reliable assessment of efficacy.[21] Participants were considered in the groups they were randomised to regardless of compliance (intention-to-treat).

## Qualitative assessment and survey of practice

In addition to the clinical assessment, semistructured qualitative interviews were undertaken with 18 patients at 6–8 weeks discharge to gauge the impact of the trial on their hospital care and eight clinical staff (four anaesthetists, one surgeon and three members of the research team) to aid insight into trial-related processes. Patients who were recruited to the trial were asked if they would be willing to be interviewed about their experiences of being in the trial. Those who consented to be interviewed were contacted by the qualitative researcher to arrange a telephone interview. The name and contact details of staff who had been involved in the trial and were willing to be interviewed were provided to the qualitative researcher, and an interview was arranged. The patient and staff telephone interviews were audiorecorded and field notes were taken in order to capture the interviewer's thoughts subsequent to completing each interview. Audiorecordings were transcribed in full by an experienced team of professional transcribers, with transcripts subsequently proof read against the recordings by the researcher. Anonymised transcripts were analysed thematically concurrently with data collection in order to allow emerging findings to be included in subsequent interviews.[22] The resulting codes and themes were refined, and consistency and variation across the interviews were explored. Once this process was completed, the resulting themes generated from the data were summarised. The main themes for patient interviews included the acceptability of the trial; motivations for being involved in the trial; experiences of participating in the trial—for example, did patients feel well informed about what involvement would entail; thoughts about the questionnaires used to assess levels of pain. The main themes from the staff interviews included experiences of patient identification and screening, experiences of the randomisation process and reflections on the trial in general terms—what processes went well and were there any areas for improvement. An electronic national survey of anaesthetists on willingness to participate in a future trial was also conducted via the Association for Cardiothoracic Anaesthesia and Critical Care membership in 26 UK thoracic units.

## Patient involvement

Our study question in chronic pain post thoracotomy is in the top-ranked research priorities voted by patients: 'What can we do to stop patients developing chronic pain after surgery?' The development of study protocol was designed in conjunction with our patient partners who also participated as a member of trial management group and trial steering committee. Our patient partners reviewed study design, paying attention to acceptability

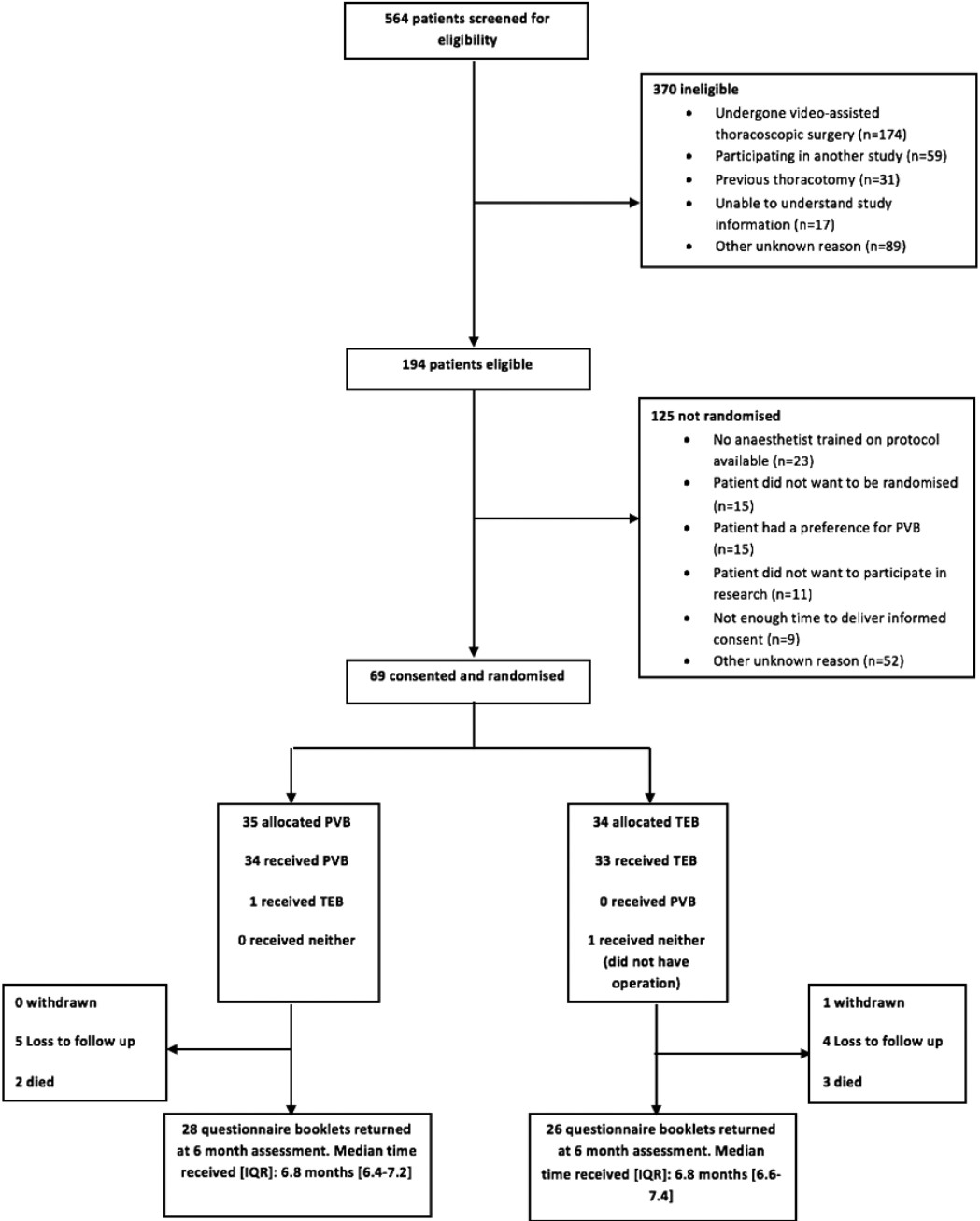

**Figure 1** Flowchart of patients through the study.

## RESULTS
### Participants and follow-up
Five hundred sixty-four patients were screened for eligibility from July 2015 to July 2016—100% of those listed for elective thoracotomy—with 194 patients found to be eligible (figure 1). The most common reasons for ineligibility were undergoing video-assisted thoracoscopic

of randomisation analgesic techniques in perioperative setting and burden of study participation. Study results including plan for future study has been disseminated within local and national patient research ambassador groups.

surgery (VATS) (174/370, 47%); participation in another study which did not allow co-enrolment (59/370, 16%) and previous thoracotomy (31/370, 8%). Of the 194 patients, 69 (36%, 95% CI 29% to 42%) were ultimately randomised. Of the others, the most common reasons for non-randomisation were anaesthetists trained to perform PVB not available (23/125, 19%); did not want to be randomised (15/125, 12%) and patient had preference for PVB (15/125, 12%). Details of randomised participants are given in table 1. The minimisation algorithm provided appropriate balance for the balancing factors; the average age of participants was 66 years with the vast majority were having a lung cancer resection operation

**Table 1** Baseline characteristics of patients

|  |  | PVB (n=35) | TEB (n=34) |
|---|---|---|---|
| Age ≥65, years* |  | 21 (60) | 20 (59) |
| Age, years | Mean (SD) | 66.0 (10.6) | 65.9 (7.9) |
| Gender=male* |  | 19 (54) | 20 (59) |
| Centre* | Birmingham Heartlands | 23 (66) | 26 (76) |
|  | University Hospital South Manchester | 12 (34) | 8 (24) |
| Reason for thoracotomy is lung cancer resection* |  | 30 (86) | 30 (88) |
| ASA physical status | 1. Normal healthy | 0 (–) | 2 (6) |
|  | 2. Mild systemic disease | 13 (37) | 10 (30) |
|  | 3. Severe systemic disease | 22 (63) | 21 (64) |
|  | Missing | – | 1 |
| ECOG performance status | 0. Normal activity | 20 (57) | 19 (58) |
|  | 1. Symptomatic but nearly fully ambulatory | 14 (40) | 14 (42) |
|  | 2. Symptomatic but bed <50% daytime | 1 (3) | 0 (–) |
|  | 3. Symptomatic bed >50% daytime | 0 (–) | 0 (–) |
|  | 4. Unable to get out of bed | 0 (–) | 0 (–) |
|  | Missing | – | 1 |
| Dyspnoea | 0. None | 17 (48) | 17 (52) |
|  | 1. Slight | 16 (46) | 13 (39) |
|  | 2. Moderate | 2 (6) | 1 (3) |
|  | 3. Moderately severe | 0 (–) | 2 (6) |
|  | 4. Severe | 0 (–) | 0 (–) |
|  | 5. Very severe | 0 (–) | 0 (–) |
|  | Missing | – | 1 |
| Smoking status | Never smoked | 7 (20) | 6 (18) |
|  | Stopped >1 year | 19 (54) | 18 (55) |
|  | Stopped >6 weeks | 3 (9) | 3 (9) |
|  | Stopped <6 weeks | 1 (3) | 4 (12) |
|  | Current smoker | 5 (14) | 2 (6) |
|  | Missing | – | 1 |
| Smoking-pack years | Median (IQR) | 37 (20, 49) | 37 (20, 50) |
|  | Missing | – | 1 |
| Alcohol units per week | Median (IQR) | 3 (0, 13.5) | 0 (0, 9.7) |
|  | Missing | – | 1 |
| Previous medical history=yes | COPD | 7 (20) | 6 (18) |
|  | Ischaemic heart disease | 4 (11) | 3 (9) |
|  | Congestive cardiac failure | 0 (–) | 1 (3) |
|  | Hypertension | 18 (51) | 15 (45) |
|  | Diabetes diet controlled | 0 (–) | 3 (9) |
|  | Diabetes oral medication | 6 (17) | 3 (9) |
|  | Diabetes insulin controlled | 2 (6) | 1 (3) |
|  | Renal failure | 1 (3) | 1 (3) |
|  | Previous stroke | 0 (–) | 0 (–) |
|  | Hyperthyroidism | 0 (–) | 0 (–) |
|  | Hypothyroidism | 5 (14) | 2 (6) |
|  | Other cancer | 12 (34) | 7 (21) |
|  | Chronic pain | 2 (6) | 5 (15) |
|  | Missing | – | 1 |
| BMI, kg/m$^2$ | Mean (SD) | 28.4 (5.8) | 27.7 (5.9) |
|  | Missing | – | 1 |

Frequency (%) presented (unless otherwise stated).
*Minimisation variable.
ASA, American Society of Anesthesiologists; BMI, body mass index; ECOG, Eastern Cooperative Oncology Group; PVB, paravertebral blockade; TEB, thoracic epidural blockade.

**Table 2** Anaesthetic technique summary

| | | PVB (n=35) | TEB (n=33)* |
|---|---|---|---|
| Difficulty in insertion | | 4 (12)† | 14 (42) |
| Failure to deliver technique | | 1 (3) | 0 (–) |
| *Intraoperative use* | | | |
| Preoperative bolus given | | 26 (74) | 25 (76) |
| Bolus drug given through catheter | | 14 (40) | 24 (73) |
| Infusion given through catheter | | 23 (66) | 21 (64) |
| *Postoperative settings* | | | |
| Local anaesthetic was Bupivacaine (as opposed to Levobupivacaine) | | 32 (94)† | 19 (58) |
| Concentration | 0.1% | 20 (59)† | 11 (33) |
| | 0.125% | 4 (12)† | 22 (67) |
| | 0.25% | 10 (29)† | 0 (–) |
| Fentanyl | None | 32 (91) | 3 (9) |
| | 2 µg/mL | 0 (–) | 0 (–) |
| | 4 µg/mL | 3 (9) | 30 (91) |
| Starting rate, mL/hour | Median (IQR) | 20 (10, 20)† | 6 (6, 8) |
| Prescribed range, mL/hour | Median (IQR) | 20 (15, 20)† | 12 (8, 15) |

Frequency (%) presented (unless otherwise stated).

*One participant received neither technique as operation was not performed for the patient.

†One response is missing (n=34).

PVB, paravertebral blockade; TEB, thoracic epidural blockade.

(60/69, 87%). Seven participants (10%) reported previous problems with (unspecified) chronic pain.

At 6 months post randomisation, 54 of 63 (86%) questionnaire booklets were returned, discounting five participants who died during the follow-up period and one participant who withdrew from the study. Of these, only one did not complete their VAS questions.

### Analgesic technique

Of the 69 randomised, 67 (97%) received their randomised intervention. One participant allocated PVB received TEB as the pleura was removed and the surgeon was unable to place the PVB. Another participant was not considered fit for surgery and was withdrawn from the study. Discounting these participants, all anaesthetic technique and intraoperative forms were completed. A summary of the anaesthetic techniques is given in table 2; a summary of the operative details and other analgesics provided to participants up to discharge is provided in the online supplementary appendix tables S1 and S2.

### Patient-reported outcomes

Levels of pain in the 3 days post operation appeared similar in both groups (see online supplementary appendix table S3). The number of patients indicating at least a moderate level of average or worst chest pain (based on commonly used threshold of >3 or ≥4[23 24]) was lower with PVB compared with TEB at 6 months, but with expected high levels of uncertainty given the limited size of sample (table 3; full results given in online supplementary appendix table S4). Results from other questionnaire responses had a lot of uncertainty (table 4).

### Safety

No concerns were expressed by the independent Oversight Committee who met twice during the recruitment period to review study progress and safety data. Complications and short-term mortality data appeared similar in both groups (see online supplementary appendix table S5). There were five deaths; only one was considered to be a serious adverse event—this was aspiration pneumonitis and was not considered related to intervention (PVB). One other participant experienced a serious adverse event in the TEB group; this participant was diagnosed with acute kidney injury in the postoperative period.

### Assessment of participant blinding

Nineteen of thirty-two (60%) patients in the TEB group correctly identified the anaesthetic technique they had been allocated at discharge and the figure for PVB was 8/31 (26%); this reduced to 13/24 (52%) in the TEB group at 6 months and 5/26 (19%) in the PVB group (see online supplementary appendix table S6).

### National survey of anaesthetists

Forty-three responses were returned. Of these, 27 anaesthetists indicated that they would be willing to randomise in a future study (63%). Five (12%) indicated that they would 'possibly' be willing to randomise with reasons of

**Table 3** Incidence of significant (>3 or ≥4) or severe pain (≥7) from Visual Analogue Scale scores

| | PVB n (%) | TEB n (%) | RR (95% CI)* |
|---|---|---|---|
| **Average chest pain overall >3** | | | |
| 3 months | 5/25 (20.0) | 6/21 (28.6) | 0.7 (0.2 to 2.0) |
| 6 months | 2/27 (7.4) | 5/25 (20.0) | 0.4 (0.1 to 1.7) |
| **Average chest pain overall ≥4** | | | |
| 3 months | 5/25 (20.0) | 4/21 (19.0) | 1.1 (0.3 to 3.4) |
| 6 months | 1/27 (3.7) | 3/25 (12.0) | 0.3 (0.03 to 2.8) |
| **Average chest pain overall ≥7** | | | |
| 3 months | 0/25 (−) | 2/21 (10.0) | −† |
| 6 months | 0/27 (−) | 1/25 (4.0) | −† |
| **Worst chest pain overall >3** | | | |
| 3 months | 9/24 (37.5) | 7/23 (30.4) | 1.2 (0.6 to 2.8) |
| 6 months | 7/27 (25.9) | 11/26 (42.3) | 0.6 (0.3 to 1.3) |
| **Worst chest pain overall ≥4** | | | |
| 3 months | 8/24 (33.3) | 7/23 (30.4) | 1.1 (0.5 to 2.5) |
| 6 months | 7/27 (25.9) | 9/26 (34.6) | 0.75 (0.3 to 1.7) |
| **Worst chest pain overall ≥7** | | | |
| 3 months | 6/24 (25.0) | 5/23 (21.7) | 1.2 (0.4 to 3.3) |
| 6 months | 2/27 (7.4) | 5/26 (19.2) | 0.4 (0.1 to 1.8) |

*RR <1 indicates less incidence with PVB.
†Unable to estimate.
PVB, paravertebral blockade; RR, relative risk; TEB, thoracic epidural blockade.

concern given as lack of skill and ward structure. The remaining 11 (26%) were unwilling.

## Qualitative interviews

Interviewed participants had a positive experience, reporting that they felt well informed by trial staff. The consent process was felt to be well undertaken, and patients felt reassured by the explanations of randomisation. A substantial number of participants reported that the study questionnaires were long and repetitive. Interviewed staff felt recruitment was successful although there were some challenges when the role was undertaken by non-clinical staff. There was a feeling that some of the data collection tools were repetitive. Clinical staff were supportive of the trial; they found that the trial processes, randomisation and procedures were very straightforward. The guidance and teaching provided for anaesthetists to get them up to speed for performing PVB was very positively received, and it was considered a relatively simple procedure to learn.

## DISCUSSION

Our four key indicators for feasibility were met, and we have shown that a large multicentre RCT of PVB versus TEB for perioperative pain control with an objective of assessing long-term post-thoracotomy chronic pain is feasible. It is possible to randomise and follow-up patients with high fidelity over 6 months. In the qualitative assessment we carried out, participants and staff were largely happy with study processes. In particular, the consent, randomisation and assessment methods were considered appropriate. As hypothesised, VAS pain scores were lower with PVB compared with TEB at 6 months on average but with high levels of uncertainty. This would need to be investigated further in a much more substantial trial.

While the evidence on short-term outcomes, for example, minor complications and analgesic efficacy, point to PVB being at least as effective as TEB,[14] current evidence on whether PVB is more effective than TEB in reducing rates of chronic pain is limited. A recent systematic review failed to attain sufficient data to compare the impact of TEB and PVB techniques on CPTP.[14] The incidence of CPTP was very poorly reported, and studies did not describe how data on chronic pain were measured or collected. The lack of evidence highlights an area in need of further research. We are not aware of any current or planned randomised trials with CPTP as an outcome that may change these findings since this systematic review was published. Incidence of CPTP was slightly lower in our study—between 30% and 34%— than previously quoted at 49%,[6] although we need to acknowledge that our figure is associated with a reasonably large amount of uncertainty given the small size of sample. A likely contributing factor to this discrepancy is the considerable and widely acknowledged variation in how pain is measured,[25] as it is dependent on the measurement instrument and threshold chosen among other factors.[23]

This feasibility work has allowed us to fine-tune study processes ahead of a larger trial. One likely modification to the future study protocol is to revise the length of the follow-up questionnaires, as the participants considered that their length was too long and repetitive in content. Shortened and simplified follow-up questionnaire would also potentially help to improve follow-up figures further, as lengthy questionnaires are a known predictor of participant drop-out from trials.[26]

Inclusion and exclusion criteria appeared appropriate and do not appear to be changed. The vast majority of ineligible participants were those scheduled to receive VATS. While this procedure is becoming more popular,[1] the comparison in question is not an appropriate one as TEB is not routinely used to provide pain relief in this type of surgery. Also, the VATS approach does not require any rib spreading, resulting in a less invasive procedure, and while chronic pain is still an issue after VATS, it is not considered to be as great as following open thoracotomy.[27] Unfortunately, while VATS is becoming more common, there will still be a substantial proportion of patients who will always be ineligible for this operation because of, for example, inability to achieve complete resection in more advanced pulmonary lung cancers. Lung cancer remains the most common cancer worldwide.[28 29]

**Table 4**  Results of patient-reported outcomes

| | PVB mean score (SD, n) | TEB mean score (SD, n) | Difference between group means (95% CI) |
|---|---|---|---|
| *BPI results summary\** | | | |
| Pain severity score (0–10, higher=worse pain) | | | |
| Baseline | 0.9 (1.7, 34) | 1.3 (2.0, 32) | −0.4 (−1.3 to 0.5) |
| 24 hours post surgery | 5.3 (1.7, 33) | 3.9 (2.7, 32) | 1.4 (0.3 to 2.5) |
| 48 hours post surgery | 5.2 (1.7, 30) | 5.1 (2.5, 31) | 0.1 (−0.9 to 1.2) |
| 72 hours post surgery | 4.9 (1.7, 23) | 5.6 (2.0, 25) | −0.7 (−1.8 to 0.4) |
| Discharge | 4.0 (1.4, 31) | 4.1 (2.4, 32) | −0.1 (−1.1 to 0.9) |
| 3 months | 1.6 (1.7, 22) | 2.5 (2.5, 21) | 0.9 (−2.2 to 0.4) |
| 6 months | 1.3 (1.8, 25) | 2.3 (2.6, 23) | −1.0 (−2.3 to 0.3) |
| Pain interference score (0–10, higher=worse pain) | | | |
| Baseline | 0.6 (1.3, 34) | 1.5 (2.2, 32) | −0.9 (−1.8 to 0.1) |
| 24 hours post surgery | 5.8 (2.1, 30) | 5.4 (2.9, 28) | 0.4 (−0.8 to 1.8) |
| 48 hours post surgery | 6.0 (2.0, 30) | 5.8 (2.2, 27) | 0.2 (−0.9 to 1.3) |
| 72 hours post surgery | 5.4 (2.3, 21) | 5.8 (2.6, 23) | −0.4 (−1.9 to 1.1) |
| Discharge | 4.7 (2.0, 23) | 4.3 (2.5, 29) | 0.4 (−0.9 to 1.7) |
| 3 months | 1.1 (1.4, 24) | 2.8 (3.0, 22) | −1.7 (−3.1 to 0.3) |
| 6 months | 1.6 (2.2, 27) | 1.9 (2.5, 25) | −0.3 (−1.6 to 1.0) |
| *NPS results summary\** | | | |
| How intense does your pain feel? (0–10, higher=worse pain) | | | |
| Baseline | 1.7 (2.8, 33) | 1.8 (2.8, 33) | −0.1 (−1.5 to 1.3) |
| Discharge | 5.8 (2.8, 31) | 5.4 (3.8, 31) | 0.4 (−1.3 to 2.1) |
| 3 months | 2.2 (2.4, 25) | 3.6 (3.8, 21) | −1.4 (−3.4 to 0.5) |
| 6 months | 2.2 (2.7, 27) | 2.7 (3.2, 26) | −0.5 (−2.2 to 1.1) |
| How sharp does your pain feel? (0–10, higher=worse pain) | | | |
| Baseline | 1.1 (2.8, 33) | 1.6 (3.4, 32) | −0.5 (−2.1 to 1.1) |
| Discharge | 5.5 (3.2, 31) | 5.4 (4.0, 31) | 0.1 (−1.7 to 1.9) |
| 3 months | 2.6 (3.0, 25) | 3.2 (4.0, 21) | −0.6 (−2.8 to 1.5) |
| 6 months | 3.0 (3.3, 27) | 2.4 (3.3, 26) | 0.6 (−1.3 to 2.4) |
| How hot does your pain feel? (0–10, higher=worse pain) | | | |
| Baseline | 1.0 (2.6, 33) | 1.3 (2.7, 32) | −0.3 (−1.6 to 1.0) |
| Discharge | 3.6 (3.4, 31) | 2.4 (2.9, 31) | 1.2 (−0.4 to 2.8) |
| 3 months | 1.6 (2.8, 25) | 2.5 (3.3, 21) | −0.9 (−2.7 to 1.0) |
| 6 months | 1.2 (2.0, 28) | 1.6 (2.9, 26) | −0.4 (−1.7 to 1.0) |
| How dull does your pain feel? (0–10, higher=worse pain) | | | |
| Baseline | 1.8 (2.9, 33) | 1.5 (2.5, 33) | 0.3 (−1.0 to 1.7) |
| Discharge | 2.9 (2.7, 30) | 3.9 (2.8, 30) | −1.0 (−2.4 to 0.5) |
| 3 months | 2.6 (2.1, 25) | 3.3 (3.3, 21) | −0.7 (−2.4 to 0.9) |
| 6 months | 2.1 (2.8, 28) | 2.3 (2.8, 26) | −0.2 (−1.8 to 1.2) |
| How cold does your pain feel? (0–10, higher=worse pain) | | | |
| Baseline | 0.3 (1.6, 33) | 0.4 (1.7, 32) | −0.1 (−0.9 to 0.7) |
| Discharge | 1.5 (2.5, 30) | 1.0 (2.2, 31) | 0.5 (−0.7 to 1.7) |
| 3 months | 0.2 (0.5, 25) | 1.5 (2.9, 21) | −1.3 (−2.7 to 0.0) |
| 6 months | 0.3 (0.9, 28) | 0.7 (1.6, 26) | −0.4 (−1.1 to 0.3) |
| How sensitive is your pain? (0–10, higher=worse pain) | | | |
| Baseline | 0.5 (1.9, 33) | 1.1 (2.5, 32) | −0.6 (−1.7 to 0.5) |
| Discharge | 2.3 (2.9, 30) | 2.8 (2.9, 32) | −0.5 (−1.9 to 1.0) |
| 3 months | 2.2 (3.0, 25) | 2.1 (2.7, 23) | 0.1 (−1.6 to 1.7) |

Continued

**Table 4** Continued

| | PVB mean score (SD, n) | TEB mean score (SD, n) | Difference between group means (95% CI) |
|---|---|---|---|
| 6 months | 1.6 (2.0, 28) | 2.5 (3.1, 26) | −0.9 (−2.4 to 0.5) |
| **How itchy is your pain? (0–10, higher=worse pain)** | | | |
| Baseline | 0 (0, 33) | 0.8 (2.0, 32) | −0.8 (−1.5 to 0.0) |
| Discharge | 2.2 (2.8, 30) | 2.6 (3.3, 31) | −0.4 (−2.0 to 1.2) |
| 3 months | 1.2 (2.5, 25) | 1.8 (2.6, 22) | −0.6 (−2.1 to 0.9) |
| 6 months | 0.4 (0.7, 28) | 2.1 (2.6, 25) | −1.7 (−2.8 to 0.6) |
| **How unpleasant is your pain? (0–10, higher=worse pain)** | | | |
| Baseline | 1.9 (2.9, 31) | 2.1 (3.3, 33) | −0.2 (−1.7 to 1.4) |
| Discharge | 6.2 (3.0, 30) | 5.6 (3.3, 30) | 0.6 (−1.1 to 2.2) |
| 3 months | 2.8 (2.6, 24) | 4.0 (3.9, 21) | −1.2 (−3.3 to 0.8) |
| 6 months | 2.4 (2.7, 28) | 3.0 (3.2, 24) | 0.6 (−2.3 to 1.0) |
| **How intense is your deep pain? (0–10, higher=worse pain)** | | | |
| Baseline | 1.3 (2.4, 32) | 2.4 (3.7, 33) | −1.1 (−2.6 to 0.5) |
| Discharge | 6.2 (2.8, 31) | 5.5 (3.4, 31) | 0.7 (−0.9 to 2.3) |
| 3 months | 3.0 (2.8, 23) | 4.1 (3.7, 21) | −1.1 (−3.1 to 0.9) |
| 6 months | 2.8 (3.0, 28) | 2.5 (2.9, 25) | 0.3 (−1.4 to 1.9) |
| **How intense is your surface pain? (0–10, higher=worse pain)** | | | |
| Baseline | 0.8 (2.1, 31) | 0.7 (1.5, 32) | 0.1 (−0.9 to 1.0) |
| Discharge | 3.9 (3.4, 30) | 3.8 (3.3, 32) | 0.1 (−1.7 to 1.7) |
| 3 months | 1.9 (2.2, 24) | 3.5 (3.6, 21) | −1.6 (−3.3 to 0.2) |
| 6 months | 1.2 (1.5, 27) | 2.7 (3.2, 24) | −1.5 (−2.9 to 0.1) |
| *EQ-5D-DL results summary†* | | | |
| EQ-5D index score (−0.59=worst outcome, 1.0=best outcome) | | | |
| Baseline | 0.75 (0.22, 35) | 0.77 (0.20, 33) | −0.02 (−0.12 to 0.08) |
| Discharge | 0.49 (0.23, 27) | 0.49 (0.23, 30) | −0.00 (−0.12 to 0.12) |
| 3 months | 0.73 (0.17, 25) | 0.65 (0.23, 23) | 0.08 (−0.04 to 0.20) |
| 6 months | 0.72 (0.23, 25) | 0.69 (0.22, 25) | 0.03 (−0.09 to 0.16) |
| EQ-5D thermometer (0–100, higher=better) | | | |
| Baseline | 76.6 (16.3, 34) | 74.3 (17.8, 33) | 2.3 (−5.9 to 10.7) |
| Discharge | 60.4 (22.7, 26) | 60.2 (20.7, 30) | 0.2 (−11.5 to 11.9) |
| 3 months | 73.2 (21.2, 25) | 67.5 (18.3, 23) | 5.7 (−5.8 to 17.2) |
| 6 months | 78.0 (14.0, 25) | 64.4 (20.4, 25) | 13.6 (3.6 to 23.6) |
| *HADS results summary‡* | | | |
| Depression index score (0–21, lower=better) | | | |
| Baseline | 2.9 (2.9, 33) | 3.0 (2.6, 32) | −0.1 (−1.4 to 1.3) |
| Discharge | 4.3 (3.4, 28) | 6.0 (5.0, 28) | −1.7 (−4.0 to 0.5) |
| 3 months | 3.6 (2.7, 25) | 5.5 (4.0, 21) | −1.9 (−4.1 to 0.1) |
| 6 months | 4.5 (3.5, 24) | 6.0 (4.6, 25) | −1.5 (−3.9 to 0.8) |
| Anxiety index score (0–21, 0-lower=better) | | | |
| Baseline | 6.3 (5.0, 34) | 6.5 (4.4, 30) | −0.2 (−2.6 to 2.1) |
| Discharge | 5.0 (4.4, 29) | 7.0 (5.3, 28) | −2.0 (−4.6 to 0.6) |
| 3 months | 5.2 (4.3, 25) | 6.1 (4.8, 19) | −0.9 (−3.7 to 1.9) |
| 6 months | 5.7 (3.6, 24) | 7.3 (5.0, 25) | −1.6 (−4.1 to 0.9) |

*Scores<0 indicate less pain with PVB.
†Scores>0 indicate better QoL with PVB.
‡Scores<0 indicate less anxiety/depression with PVB.
BPI, brief pain inventory; HADS, hospital anxiety and depression index score; NPS, Neuropathic Pain Scale; PVB, paravertebral blockade; QoL, quality of life; TEB, thoracic epidural blockade.

It was apparent from the assessment of the ability to blind the participant to the analgesic technique that complete blinding is unlikely to be possible. Substantially, more participants appeared to be able to determine they had had an epidural blockade than the paravertebral technique. Attempts to completely blind participants in any future study are likely to make processes prohibitively complex and expensive due to innate differences in the mode of delivery and action in the two analgesic techniques. This could include sham insertion techniques to disguise the randomised intervention. The question is whether patient blinding is likely to be an important potential cause of detection bias, unduly affecting the results of any future trial. The proposed primary outcome of pain rating is subjective in nature, and this is a theoretical possibility. However, we believe that there is no reason to suspect that recipients of the randomised interventions have strong preconceptions with regard to the relative effectiveness of each analgesic technique. Furthermore, the primary outcome would be collected via questionnaires administered by post or phone, at a time remote from the original operative procedure, which are likely to be resilient to the effects of imperfect concealment.

Findings from the survey of practice were encouraging; while we set a target of 70% of thoracic anaesthetists willing to randomise to a future trial as being indicative of success, we found 63% willing and 12% 'possibly willing'. To achieve 70%, we would require solutions for those anaesthetists expressing uncertainty. The major reasons given were lack of skill and ward structure. These hurdles do not appear insurmountable if funding for further training and extra nurse support could be provided; something a large funding body may be willing to provide. A successful trial would also require full engagement with anaesthetic and thoracic surgical trainees' network to maximise recruitment.

We have reported the study to randomised pilot study standards as recommended by Consolidated Standards of Reporting Trials extension for pilot study.[30] Also, as recommended,[31] we set objectives a priori to determine whether the pilot study would demonstrate feasibility of a larger trial. Since this study was not a definitive trial, we have been careful not to overinterpret the findings. Hence at this stage, we cannot expect the outcomes of this pilot study to be directly translated into clinical care, as the study was not large enough to be able to detect small-to-moderate realistic-sized difference in rates of chronic pain, nor was the scope wide enough in terms of the number of centres involved to return a generalisable result.

Nevertheless, this pilot study is an important precursor to a larger, substantive, trial and provides invaluable information that will help to ensure its success. A definitive study—instigated by the authors of this manuscript—has now been funded by the National Institute of Health Research (NIHR) Health Technology Assessment Programme and is expected to start recruiting in 2018 from 20 UK thoracic centres. The design of this study has been partially informed by the results of this pilot. The primary outcome will be incidence of chronic pain at 6 months attained from VAS responses similar to that used here; other outcomes will follow Initiative on Methods, Measurement, and Pain Assessment in Clinical Trials (IMMPACT) recommendations.[32] A total sample size of 1026 participants will give 90% power (p=0.05) to detect a reduction in incidence of CPTP of 30% with TEB (a similar event rate to that noted in this pilot study) down to 20% with PVB. The rate of recruitment noted in this study has been factored into the number of sites and length of the recruitment period required; we estimate access to a pool of 5000 participants over a two-and-a-half year recruitment period and need to recruit a fifth of these. The design will allow a definitive answer to this important question.

**Author affiliations**
[1]Warwick Medical School, University of Warwick, Coventry, UK
[2]Academic Department of Anaesthesia, Critical Care, Pain and Resuscitation, Birmingham Heartlands Hospital, University Hospital Birmingham NHS Foundation Trust, Birmingham, UK
[3]School of Health and Population Sciences, University of Birmingham, Birmingham, UK
[4]Birmingham Clinical Trials Unit, University of Birmingham, Birmingham, UK
[5]Department of Thoracic Surgery, Birmingham Heartlands Hospital, University Hospital Birmingham NHS Foundation Trust, Birmingham, UK
[6]Nottingham Clinical Trials Unit, University of Nottingham, Nottingham, UK
[7]Institute of Inflammation and Ageing, College of Medical and Dental Sciences, University of Birmingham, Birmingham, UK
[8]University of Liverpool, Institute of Translational Medicine, Liverpool, UK
[9]Health Services Research, University of Sheffield, Sheffield, UK
[10]Department of Anaesthesia, University Hospital of South Manchester NHS Foundation Trust, Manchester, UK
[11]Institute of Applied Health Research, University of Birmingham, Birmingham, UK
[12]Department of Thoracic Surgery, University Hospital of South Manchester NHS Foundation Trust, Manchester, UK

**Acknowledgements** We thank the trial participants; members of the trial Oversight Committee: Tom Treasure (Chair), Alistair Macfie (independent anaesthetist), Mahmoud Loubani (independent surgeon), Lucy Bradshaw (independent statistician), Alistair Murray (patient and public representative); the Clinical Research Ambassador Group at Heart of England NHS Foundation Trust; Birmingham Clinical Trials Unit staff: Matt Hill (database programmer) and Hannah Watson (data manager).

**Contributors** FG, JY, JD and MW conceived the topic. FG, JY, JD, LM and AG designed the study with input from SK, LS, BN, MW, TM and AW. FG oversaw the running of the trial and all the authors contributed to the ongoing management of the trial. AK, RS and BN collected data for the trial. KT performed the statistical analysis. SF evaluated the qualitative data. The manuscript was drafted by LM, JY and FG with contributions from the other authors. All the authors contributed to the interpretation of the output and revised and reviewed the paper; they are the guarantors. The Birmingham Clinical Trials Unit undertook the randomisation and data management and monitoring. The authors had full access to all the data from the study. The authors vouch for the accuracy and completeness of the data and analyses. All authors read and approved the final manuscript.

**Funding** The study was funded by a grant from the National Institute for Health Research for Patient Benefit (RFPB grant number PB-PG-0213-30126). The views expressed are those of the author(s) and not necessarily those of the NIHR or the Department of Health and Social Care. SF is funded by the National Institute of Health Research (NIHR) Collaboration for Leadership in Applied Health Research and Care West Midlands (CLAHRCWM). The study sponsor was the Heart of England NHS Foundation Trust.

**Disclaimer** The funder had no role in study design, data collection, interpretation or analysis or in writing the report for publication.

**Competing interests** None declared.

**Patient consent for publication** Not required.

**Ethics approval** This study was approved by NHS Research Ethics Committee (NRES Committee East Midlands - Derby, REC number 14/EM/1280) and was prospectively registered (ISRCTN45041624).

**Provenance and peer review** Not commissioned; externally peer reviewed.

**Data sharing statement** All available study data can be obtained from the corresponding author.

**Open access** This is an open access article distributed in accordance with the Creative Commons Attribution 4.0 Unported (CC BY 4.0) license, which permits others to copy, redistribute, remix, transform and build upon this work for any purpose, provided the original work is properly cited, a link to the licence is given, and indication of whether changes were made. See: https://creativecommons.org/licenses/by/4.0/.

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
