## [Reviewer comments · BMJ Open]

ARTICLE DETAILS

TITLE (PROVISIONAL)	A RANDOMISED CONTROLLED TRIAL TO INVESTIGATE THE EFFECTIVENESS OF THORACIC EPIDURAL AND PARAVERTEBRAL BLOCKADE IN REDUCING CHRONIC POST-THORACOTOMY PAIN (TOPIC) – A PILOT STUDY TO ASSESS FEASIBILITY OF A LARGE MULTI-CENTRE TRIAL
AUTHORS	Yeung, Joyce; Middleton, Lee; Tryposkiadis, Kostas; Kerr, Amy; Daniels, Jane; Naidu, Babu; Melody, Teresa; Goebel, Andreas; Wilson, Matthew; Kumar, Sajith; Szentgyorgyi, Lajos; Flanagan, Sarah; Shah, Rajesh; Worrall, Antony; Gao, Fang

VERSION 1 – REVIEW

REVIEWER	Sujeet Kumar Singh Gautam Sanjay Gandhi Post-graduate Institute of Medical Sciences, Lucknow, India
REVIEW RETURNED	07-May-2018

GENERAL COMMENTS	How the authors would explain the use of VAS scale by postal questionnaire?
---

REVIEWER	Hilde M. Norum Department of Research and Development, Division of Emergencies and Critical Care, Oslo University Hospital Rikshospitalet, Norway
REVIEW RETURNED	15-May-2018

GENERAL COMMENTS	Yeung and colleagues have undertaken a pilot study to assess feasibility of a large multi-centre trial on effectiveness of thoracic epidural blockade (TEB) versus thoracic paravertebral blockade (PVB) in reducing chronic post-thoracotomy pain (CTPP). The topic for the definitive multi-centre study is highly clinically relevant and scientifically interesting, but important aspects may be improved. Concerning background: PVB is also an epidural block (Purcell-Jones, Pither et al. 1989, Karmakar, Kwok et al. 2000, Luyet, Eichenberger et al. 2009, Cowie, McGlade et al. 2010), which should be mentioned and discussed, confer doses of analgesics to be applied and safety concerns in the study. Concerning exclusion criteria: Guidelines for patients receiving antithrombotic or thrombolytic therapy state "For patients undergoing perineuraxial, deep plexus, or deep peripheral block, we recommend that guidelines regarding neuraxial techniques be similarly applied"(Horlocker, Vandermeulen et al. 2018). Whether the protocol adheres to the guidelines or not, is not reported in the present manuscript, and should be mentioned and explained.
---

	Concerning interventions: The two component drug solution for use in the thoracic epidural catheter may be changed for an analgesically more efficient triple component, low dose drug solution containing a local anaesthetic, a lipophilic opioid and a adrenergic agonist to provide pain relief with a minimum of side effects (H. 1995, Niemi and Breivik 2002, Niemi and Breivik 2003). Comparing the two methods for post-thoracotomy pain relief is more informative when each is performed according to best clinical practice. Also, the blocks should be tested with ice cubes before the patients are put under general anesthesia. Concerning clinical and participant reported measures: Total dose of analgesics administered in the TEB and PVB plus duration of infusion should be reported. Also, signs of local anesthetic systemic toxicity (LAST) should be reported, confer the risk of local anaesthetic accumulation when high doses are applied to provide PVB based pain relief (Richardson, Sabanathan et al. 1999). Further, it could be suggested to measure concentrations of local anaesthetics, to possibly address the concerns of LAST associated with PVB. Also, the authors should describe how VAS was scored -were the patients asked to mark pain intensity on a 10 cm line? Table 1: Too many lines make reduces the readability of the table. Table 2: Total doses of local anaesthetic and opioid should be reported. Duration of infusions should be reported. Table 4: With 6 out of 70 reported scores having 95% CI not including 0 for difference between the groups, the sentence "Similar patterns were seen from other questionnaire responses, with scores generally favouring PVB on average, although most estimates of uncertainty did not discount parity" may be rephrased to highlight the uncertainty, which is the most substantial finding. Also, the table is hard to read and may be redesigned. Concerning Qualitative interviews: In the manuscript, I do not find the data underlying this paragraph. Could there be a misunderstanding? General comment: The four research questions for pilot outcomes are answered by the authors in the discussion. I still have some comments. I am impressed by the high percentage (84% of those still alive) of participants returning questionnaire booklets at six months, and support the suggested shortening of the follow-up questionnaires. The National Survey of anaesthetists was carried out in 26 thoracic units, and received 43 responses, out of which 27, just about one anaesthetist per unit, indicated willingness to randomise for the future trial. This may prove to be a challenge for the future trial, with a substantial risk of some units not participating at all, and I do not fully share the authors' optimism when they describe the result of the National Survey as encouraging. The authors state in the discussion that they will not over-interpret their findings. Nevertheless, they report that "VAS pain scores were lower with PVB compared with TEB at six months on average, ..". More patients in the TEB group than in the PVB group had chronic pain prior to surgery, which may be commented on (Kehlet, Jensen et al. 2006). The lack of reported total doses of drugs administered via PVB or TEB further weakens the report, including brief "safety" paragraph. The authors discuss future blinding of the participants to the analgesic technique, and conclude that they "have no reason to suspect that recipients of the randomised interventions have strong pre-conceptions with regard to the relative effectiveness of each analgesic technique". This is somewhat contradictory to their own report on patient
--	--

preference for PVB as reason for non-randomisation (12%). Thus, in the future trial, attempts of blinding should be encouraged. In the present study, 60 procedures were carried out during a 12 month period, out of which 12 PVBs and 8 TEBs in the smaller contributing unit, reflecting one PVB per month and one TEB per more than six weeks. For 40% of the TEBs, difficulty in insertion was reported, emphasizing the need for experienced anaesthetists to do the procedure (possibly with the assistance of ultrasound?), and should be commented on by the authors. The future trial will take place in 20 thoracic units and include 1000 participants (one fifth of 5000) during a 30 month period. If the units are of equal size, which is rather improbable, each unit will include on the average 1.67 patients each month. With a 1:1 randomisation, each unit will on the average have less than one patient for each study arm per month. This could imply concerns of lack of continuity and need for advanced statistics addressing heterogeneity, which should be discussed in the present manuscript.

Cowie, B., D. McGlade, J. Ivanusic and M. J. Barrington (2010). "Ultrasound-guided thoracic paravertebral blockade: a cadaveric study." *Anesth Analg* 110(6): 1735-1739.

H., B. (1995). "Safe perioperative spinal and epidural analgesia: importance of drug combinations, segmental site of injection, training and monitoring." *Acta Anaesthesiologica Scandinavica* 39(7): 869-871.

Horlocker, T. T., E. Vandermeulen, S. L. Kopp, W. Gogarten, L. R. Leffert and H. T. Benzon (2018). "Regional Anesthesia in the Patient Receiving Antithrombotic or Thrombolytic Therapy: American Society of Regional Anesthesia and Pain Medicine Evidence-Based Guidelines (Fourth Edition)." *Reg Anesth Pain Med* 43(3): 263-309.

Karmakar, M. K., W. H. Kwok and J. Kew (2000). "Thoracic paravertebral block: radiological evidence of contralateral spread anterior to the vertebral bodies." *British Journal of Anaesthesia* 84(2): 263-265.

Kehlet, H., T. S. Jensen and C. J. Woolf (2006). "Persistent postsurgical pain: risk factors and prevention." *Lancet* 367(9522): 1618-1625.

Luyet, C., U. Eichenberger, R. Greif, A. Vogt, Z. Szucs Farkas and B. Moriggi (2009). "Ultrasound-guided paravertebral puncture and placement of catheters in human cadavers: an imaging study." *Br J Anaesth* 102(4): 534-539.

Niemi, G. and H. Breivik (2002). "Epinephrine markedly improves thoracic epidural analgesia produced by a small-dose infusion of ropivacaine, fentanyl, and epinephrine after major thoracic or abdominal surgery: a randomized, double-blinded crossover study with and without epinephrine." *Anesth Analg* 94(6): 1598-1605, table of contents.

Niemi, G. and H. Breivik (2003). "The minimally effective concentration of adrenaline in a low-concentration thoracic epidural analgesic infusion of bupivacaine, fentanyl and adrenaline after major surgery. A randomized, double-blind, dose-finding study." *Acta Anaesthesiol Scand* 47(4): 439-450.

Purcell-Jones, G., C. E. Pither and D. M. Justins (1989). "Paravertebral somatic nerve block: a clinical, radiographic, and computed tomographic study in chronic pain patients." *Anesth Analg* 68(1): 32-39.

Richardson, J., S. Sabanathan, J. Jones, R. D. Shah, S. Cheema and A. J. Mearns (1999). "A prospective, randomized comparison of preoperative and continuous balanced epidural or paravertebral

	bupivacaine on post-thoracotomy pain, pulmonary function and stress responses." Br J Anaesth 83(3): 387-392.
REVIEWER	Colin McCartney University of Ottawa, ON, Canada
REVIEW RETURNED	26-May-2018
GENERAL COMMENTS	A nicely written manuscript and clearly this has achieved funding priority. I would like to see more discussion around the primary outcome measure and power calculation earlier in the manuscript if possible.

VERSION 1 – AUTHOR RESPONSE

Responses to reviewer's comments

Reviewer 1		
How the authors would explain the use of VAS scale by postal questionnaire?	All patients have already completed identical questionnaires during in-hospital stay with research nurse support. The follow-up questionnaires also had instructions to support patients.	No changes made.
Reviewer 2		
Concerning background: PVB is also an epidural block (Purcell-Jones, Pither et al. 1989, Karmakar, Kwok et al. 2000, Luyet, Eichenberger et al. 2009, Cowie, McGlade et al. 2010), which should be mentioned and discussed, confer doses of analgesics to be applied and safety concerns in the study.	We disagree with the reviewer that PVB is also an epidural block. However we do acknowledge that PVB injection can spread beyond PVB space. We have added a sentence to describe potential spread of PVB injection. Of the studies that have been quoted by reviewer: Purcell-Jones, Pither et al. 1989 – 5ml injection but standard technique was not described; Karmakar, Kwok et al. 2000 was a single case report; Luyet, Eichenberger et al. 2009 and Cowie, McGlade et al. 2010 were cadaveric studies.	Sentence has been inserted to background. 'There are studies which describe the spread of PVB injections anteriorly across the heads and necks of the ribs to the spaces above and below; medially through an intervertebral foramen or spread laterally in the intercostal plane.' ⁹

Concerning exclusion criteria: Guidelines for patients receiving antithrombotic or thrombolytic therapy state “For patients undergoing perineuraxial, deep plexus, or deep peripheral block, we recommend that guidelines regarding neuraxial techniques be similarly applied”(Horlocker, Vandermeuelen et al. 2018). Whether the protocol adheres to the guidelines or not, is not reported in the present manuscript, and should be mentioned and explained.	The performance of both techniques and exclusion criteria relating to coagulation abnormalities adhere to national guidelines (Association of Anaesthetists of Great Britain and Ireland 2013).	The sentence has been changed to specify national guidelines: ‘coagulation abnormalities (in accordance with Association of Anaesthetists in Great Britain and Ireland guidance 2013)’
Concerning interventions: The two component drug solution for use in the thoracic epidural catheter may be changed for an analgesically more efficient triple component, low dose drug solution containing a local anaesthetic, a lipophilic opioid and a adrenergic agonist to provide pain relief with a minimum of side effects (H. 1995, Niemi and Breivik 2002, Niemi and Breivik 2003). Comparing the two methods for post-thoracotomy pain relief is more informative when each is performed according to best clinical practice. Also, the blocks should be tested with ice cubes before the patients are put under general anesthesia.	TEB and PVB protocols were designed in conjunction with thoracic anaesthetists and surgeons. The technique and the choice of drug solution closely followed TOPIC pilot study was designed to reflect current UK practice and this differ from what the reviewer has suggested. Anaesthetists would test for the effectiveness of the block as their usual practice and this is not mandated in the study protocol. Testing with ice cubes is not routine practice in UK anaesthetic practice.	No changes made.
Concerning clinical and participant reported measures: Total dose of analgesics administered in the TEB and PVB plus duration of infusion should	TOPIC pilot study was a feasibility study to assess patient recruitment into a randomised study comparing TEB and PVB and measuring chronic pain at 6 months. The dose (infusion, total dose) was not recorded in pilot	No change required.

be reported. Also, signs of local anesthetic systemic toxicity (LAST) should be reported, confer the risk of local anaesthetic accumulation when high doses are applied to provide PVB based pain relief (Richardson, Sabanathan et al. 1999). Further, it could be suggested to measure concentrations of local anaesthetics, to possibly address the concerns of LAST associated with PVB.	study, however, this data will be captured in large definitive funded study and should satisfy what the reviewer would like to see (study starting in Autumn 2018). LA toxicity was collected as part of adverse events/serious adverse events. None was reported in this pilot study.	
Also, the authors should describe how VAS was scored -were the patients asked to mark pain intensity on a 10 cm line?	Patients were asked to mark on a 10cm line as shown below 	Description has been added to the sentence: Visual Analogue Scales (VAS)[16] for worst and average chest pain (overall, at rest, after coughing, after moving, after physiotherapy; with higher score - maximum 10 - indicating higher levels of pain, as marked along a 10cm line),
Table 1: Too many lines make reduces the readability of the table.	If the manuscript is accepted, we will work with the journal editors to provide a format that matches the in-house style.	No changes made.
Table 2: Total doses of local anaesthetic and opioid should be reported. Duration of infusions should be reported.	TOPIC pilot study was a feasibility study to assess patient recruitment into a randomised study comparing TEB and PVB and measuring chronic pain at 6 months. As TOPIC pilot is not an effectiveness study, the dose (infusion, total dose) was not recorded, however, this data will be captured in large definitive funded study and should satisfy what the reviewer would like to see (study starting in Autumn 2018).	No changes made.
Table 4: With 6 out of 70 reported scores having 95% CI not including 0 for difference between the groups, the sentence	Thanks for this comment, we agree and have edited the sentence as per the suggestion.	Instead of “Similar patterns were seen from other questionnaire responses, with scores generally favouring PVB on average,

“Similar patterns were seen from other questionnaire responses, with scores generally favouring PVB on average, although most estimates of uncertainty did not discount parity” may be rephrased to highlight the uncertainty, which is the most substantial finding. Also, the table is hard to read and may be redesigned.	Regarding the format of the table. We believe this is a simple line listing of results by questionnaire and time-point. We will work with the journal editors to provide a format that matches the in-house style if they think this is necessary.	although most estimates of uncertainty did not discount parity and there was substantial uncertainty (Table 4).”, it now reads: “Results from other questionnaire responses had a lot of uncertainty (Table 4).”
Concerning Qualitative interviews: In the manuscript, I do not find the data underlying this paragraph. Could there be a misunderstanding?	Qualitative interviews were conducted as part of process evaluation. Results were given in the manuscript: ‘Interviewed participants had a positive experience, reporting that they felt well informed by trial staff. The consent process was felt to be well undertaken and patients felt reassured by the explanations of randomisation. A substantial number of participants reported that the study questionnaires were long and repetitive. Interviewed staff felt recruitment was successful although there were some challenges when the role was undertaken by non-clinical staff. There was a feeling that some of the data collection tools were repetitive. Clinical staff were supportive of the trial; they found that the trial processes, randomisation and procedures were very straightforward. The guidance and teaching provided for anaesthetists to get them up to speed for performing PVB was very positively received and it was considered a relatively simple procedure to learn.’	No changes required.
General comment: The four research questions for pilot outcomes are answered by the authors in the discussion. I still have	Thank you for your comment. The definitive study has the support of both thoracic anaesthetists and surgeons. The study has also been adopted by UK Perioperative Clinical Trials Network and received very	No changes made.

some comments. I am impressed by the high percentage (84% of those still alive) of participants returning questionnaire booklets at six months, and support the suggested shortening of the follow-up questionnaires. The National Survey of anaesthetists was carried out in 26 thoracic units, and received 43 responses, out of which 27, just about one anaesthetist per unit, indicated willingness to randomise for the future trial. This may prove to be a challenge for the future trial, with a substantial risk of some units not participating at all, and I do not fully share the authors' optimism when they describe the result of the National Survey as encouraging.	positive feedback from Society of Thoracic Surgery at their recent meeting. Furthermore, from the experience of Birmingham Clinical Trials Unit and TOPIC study management group in delivering multicentre randomised controlled trials, the authors believe that the findings of the feasibility study and the national survey provides a strong starting point for the full study.	
The authors state in the discussion that they will not over-interpret their findings. Nevertheless, they report that "VAS pain scores were lower with PVB compared with TEB at six months on average, ..". More patients in the TEB group than in the PVB group had chronic pain prior to surgery, which may be commented on (Kehlet, Jensen et al. 2006).	As TOPIC pilot study does not assess clinical effectiveness, the authors believe that the sentence 'VAS pain scores were lower with PVB compared with TEB at six months on average but with high levels of uncertainty' provides a balanced view. The discussion also specifically how firm conclusions should not be drawn and results are required from a definitive study.	No changes made.
The lack of reported total doses of drugs administered via PVB or TEB further weakens the report, including brief "safety" paragraph.	In this feasibility study report, the authors have included all the two serious adverse events. Despite the lack of data on total doses administered, consultant anaesthetists participating in this study would be following TOPIC intervention protocol with drug and infusion doses (published Yeung J et al. Randomised controlled pilot study	No changes made.

	to investigate the effectiveness of thoracic epidural and paravertebral blockade in reducing chronic post-thoracotomy pain: TOPIC feasibility study protocol. BMJ Open. 2016 Dec 1;6(12):e012735)	
The authors discuss future blinding of the participants to the analgesic technique, and conclude that they “have no reason to suspect that recipients of the randomised interventions have strong pre-conceptions with regard to the relative effectiveness of each analgesic technique”. This is somewhat contradictory to their own report on patient preference for PVB as reason for non-randomisation (12%). Thus, in the future trial, attempts of blinding should be encouraged.	The authors do not believe the lack of blinding of patients would influence the reporting of chronic pain at 6 and 12 months post-surgery (outcomes in definitive study). Patient preference of a technique represent a reason to decline study participation and will not be affected by blinding of group allocation. Other reasons for declining consent which included ‘did not want to be randomised (15/125, 12%); patient had preference for PVB (15/125, 12%)’. The proportion of these patients was small and did not impact on overall recruitment target.	No changes made
In the present study, 60 procedures were carried out during a 12 month period, out of which 12 PVBs and 8 TEBs in the smaller contributing unit, reflecting one PVB per month and one TEB per more than six weeks. For 40% of the TEBs, difficulty in insertion was reported, emphasizing the need for experienced anaesthetists to do the procedure (possibly with the assistance of ultrasound?), and should be commented on by the authors. The future trial will take place in 20 thoracic units and include 1000 participants (one fifth of	Thank you for the reviewer’s comments. In the definitive study protocol, it is a requirement that the participating anaesthetists will be consultants and senior trainees who can perform both PVB and TEB competently and to expert level. In a large randomised controlled trial setting, there will be centres who can recruit more patients per month and smaller centres who recruit fewer patients. The demonstrated feasibility of recruiting patients from two medium size centres meant that the study team is confidently that patient recruitment in definitive study is achievable. As this manuscript is the report of the feasibility study and the word limit, the authors do not feel that detailed	No changes made.

5000) during a 30 month period. If the units are of equal size, which is rather improbable, each unit will include on the average 1.67 patients each month. With a 1:1 randomisation, each unit will on the average have less than one patient for each study arm per month. This could imply concerns of lack of continuity and need for advanced statistics addressing heterogeneity, which should be discussed in the present manuscript.	discussion of the definitive study (which will subsequently be published) is warranted.	
Reviewer 3		
A nicely written manuscript and clearly this has achieved funding priority. I would like to see more discussion around the primary outcome measure and power calculation earlier in the manuscript if possible.	Thank you for your comments. The structure of the manuscript is in accordance with the Journal instructions and also reporting guidance set out by CONSORT.	No changes made

VERSION 2 – REVIEW

REVIEWER	Hilde M. Norum Department of Research and Development, Division of Emergencies and Critical Care, Oslo University Hospital Rikshospitalet, Norway
REVIEW RETURNED	22-Aug-2018

GENERAL COMMENTS	Yeung and colleagues have undertaken a pilot study to assess feasibility of a large multi-centre trial on effectiveness of thoracic epidural blockade (TEB) versus thoracic paravertebral blockade (PVB) in reducing chronic post-thoracotomy pain (CTPP), and revised their manuscript. I thank the authors and the editors for the opportunity to review the revised manuscript R1. For most of my concerns, the authors have responded to my full satisfaction; elucidating and with appropriate manuscript actions, and I have only a very few comments to R1.
---

	Concerning the abstract: This is a study of feasibility, which is important and could be clearly stated in the “objectives” paragraph of the abstract. Concerning background: Informatively, the authors have included a sentence on spread of PVB injections. Dura extends into the paravertebral space (Boezaart, Lucas et al. 2009), which may be described by the authors. This would add clarity without conflicting with the authors’ disagreement with my conception of PVB being also an epidural block. Concerning Qualitative interviews: In their reply, the authors state that ‘Qualitative interviews were conducted as part of process evaluation. Results were given in the manuscript’ and ‘No actions required’. Unfortunately, in neither the METHODS part nor the RESULTS part of R1, a description of the principles and procedures for data organization and analysis of the data collected in the semistructured qualitative interviews is included. The authors should allow the readers of their manuscript to understand what happened to the raw material to arrive at the results (Malterud 2001), for instance by incorporating a brief explanation. References Boezaart, A. P., S. D. Lucas and C. E. Elliott (2009). "Paravertebral block: cervical, thoracic, lumbar, and sacral." Current Opinion in Anesthesiology 22(5): 637-643. Malterud, K. (2001). "Qualitative research: standards, challenges, and guidelines." Lancet 358(9280): 483-488.
--	--

VERSION 2 – AUTHOR RESPONSE

Reviewer 1 Dr Hilde M. Norum’s comments	Responses	Action
Concerning the abstract: This is a study of feasibility, which is important and could be clearly stated in the “objectives” paragraph of the abstract.	We have added this to the objectives in abstract.	The specific objective of this pilot study was to assess the feasibility of conducting a larger trial to determine whether PVB at thoracotomy is more effective in reducing CPTP compared to TEB.
Concerning background: Informatively, the authors have included a sentence on spread of PVB injections. Dura extends into the paravertebral space (Boezaart, Lucas et al. 2009), which may be described by the authors. This would add clarity without conflicting with the authors’ disagreement with my	We understand the reviewer’s viewpoint that the paravertebral space can communicate with the epidural space and that high volume PVB block can spread into the epidural space. The description of the TEB and PVB and the boundary of the paravertebral space provided by the authors merely serves as an	We added the following description of paravertebral space to address reviewer’s concerns. Paravertebral blockade (PVB), involves injecting local anaesthetic into the paravertebral space, which contains spinal nerves (and sometimes even extension of the dura), white and grey rami communicantes, the sympathetic chain and

conception of PVB being also an epidural block.	introduction and background of the two techniques. We would aim to include these wider discussions in the publication of our funded definitive study and its effects on chronic pain, if any.	intercostal vessels, on the side of surgery. ⁹
Concerning Qualitative interviews: In their reply, the authors state that ‘Qualitative interviews were conducted as part of process evaluation. Results were given in the manuscript’ and ‘No actions required’. Unfortunately, in neither the METHODS part nor the RESULTS part of R1, a description of the principles and procedures for data organization and analysis of the data collected in the semistructured qualitative interviews is included. The authors should allow the readers of their manuscript to understand what happened to the raw material to arrive at the results (Malterud 2001), for instance by incorporating a brief explanation.	We have added this to the manuscript under Qualitative interviews and survey of practice section (page 7).	The patient and staff telephone interviews were audio-recorded and field notes were be taken to in order to capture the interviewer’s thoughts subsequent to completing each interview. Audio-recordings were transcribed in full by an experienced team of professional transcribers, with transcripts subsequently proof read against the recordings by the researcher. Anonymised transcripts were analysed thematically concurrently with data collection in order to allow emerging findings to be included in subsequent interviews.²² The resulting codes and themes were refined and consistency and variation across the interviews was explored. Once this process was completed, the resulting themes generated from the data were summarised.

VERSION 3 – REVIEW

REVIEWER	Hilde M. Norum Department of Research and Development, Division of Emergencies and Critical Care, Oslo University Hospital, Oslo, Norway
REVIEW RETURNED	20-Nov-2018
GENERAL COMMENTS	A RANDOMISED CONTROLLED TRIAL TO INVESTIGATE THE EFFECTIVENESS OF THORACIC EPIDURAL AND PARAVERTEBRAL BLOCKADE IN REDUCING CHRONIC POST-THORACOTOMY PAIN (TOPIC) – A PILOT STUDY TO ASSESS FEASIBILITY OF A LARGE MULTI-CENTRE TRIAL, R2

	Comments to the authors I thank the authors and the editor for allowing me to review the revised manuscript. The authors have addressed the questions I raised concerning the preceding manuscript by including a response table and incorporating their responses in the revised manuscript. I think the included responses are adequate, except for the response to my comment on the semi-structured qualitative interviews. The response nicely describes in more detail how the interviews were done, but not which thematic issues were covered, and how the participants were recruited. Thus, it is still challenging for the reader to understand how the results emerged and consequently, for the reader to benefit from the discussion. Concerning the abstract: The abstract in the manuscript (Objectives: Thoracotomy is considered one of the most painful surgical procedures. The incidence of chronic post-thoracotomy pain (CPTP) is up to 50%. Paravertebral blockade (PVB) may be superior to thoracic epidural blockade (TEB) in preventing CPTP. The specific objective of this pilot study was to assess the feasibility of conducting a larger trial to determine whether PVB at thoracotomy is more effective in reducing CPTP compared to TEB. Design: A randomised, parallel, external pilot study was conducted to assess whether a large randomised trial of TEB and PVB with CPTP as the primary outcome is feasible. Setting: Two adult thoracic centres in the UK Participants: All adult patients admitted for elective open thoracotomy. Participants were excluded if they were ASA physical status IV or V; or if there is contraindication to local anaesthetics; infection near the proposed puncture site; coagulation/thoracic spine disorders; required chest wall resection or emergency thoracic surgery or had a previous thoracotomy. Results: All patients presenting for thoracotomy were screened over a 12-month period with 194 found to be eligible. Of these, 69 (36%) were randomised (95%CI: 29%-42%). Discounting five participants who died, 54 of 64 participants (84%) returned questionnaire booklets at six months. The number of participants indicating at least a moderate level of chest pain at six months was lower with PVB but with high levels of uncertainty (RR: 0.7; 95%CI: 0.3-1.7 for worst pain; RR: 0.3; 95%CI: 0.0-2.8 for average). There were no safety concerns. Conclusions: A large multicentre RCT of PVB versus TEB is feasible as it is possible to randomise and follow up participants with high fidelity. Pain scores were lower on average with PVB compared to TEB but a much larger trial is required to confirm this reliably.) differs from the one included online: Abstract bmjopen-2018-023679.R2 Background: Thoracotomy is considered one of the most painful surgical procedures. The incidence of chronic post-thoracotomy pain (CPTP) is up to 50%. Paravertebral blockade (PVB) may be superior to thoracic epidural blockade (TEB) in preventing CPTP. We wanted to assess whether a large randomised trial of these two peri-operative analgesic techniques with CPTP as the primary outcome is feasible. Methods: In this randomised, parallel, external pilot trial in two adult thoracic centres in the UK, patients admitted for elective thoracotomy were randomised to either TEB or PVB. The main feasibility outcomes were recruitment rate, screening rate and data completion. Participant-reported outcomes included worst and average chest pain scores on a visual analogue scale at
--	--

	six months post-randomisation. Results: All patients presenting for thoracotomy were screened over a 12-month period with 194 found to be eligible. Of these, 69 (36%) were randomised (95%CI: 29%-42%). Discounting five participants who died, 54 of 64 participants (84%) returned questionnaire booklets at six months. The number of participants indicating at least a moderate level of chest pain at six months was lower with PVB but with high levels of uncertainty (RR: 0.7; 95%CI: 0.3-1.7 for worst pain; RR: 0.3; 95%CI: 0.0-2.8 for average). There were no safety concerns. Conclusions: A large multicentre RCT of PVB versus TEB is feasible as it is possible to randomise and follow up participants with high fidelity. Pain scores were lower on average with PVB compared to TEB but a much larger trial is required to confirm this reliably. Given by this difference, I had to choose one of the abstracts to be able to revise the manuscript. I have based my revision on the abstract included in the revised manuscript, not the one presented online. For the original manuscript, I commented on several tables, and as I understood from the authors' response and the first revised version of the manuscript, the tables would be revised for the final version of the manuscript, which I anticipate would be the present. Therefore, I allow myself to comment on the tables in the current, second revised version of the manuscript. Please ignore my comments to the tables if I have misunderstood concerning the status of the tables. The paragraph "Participants and follow-up" and the included "Table 1. Baseline characteristics of patients" lack a statistical comparison of the two included study groups -were the groups comparable? This should be included in the text and/ or by a p-value in the table. Concerning "Table 2. Anaesthetic technique summary": The explanation for superscript 1, "1 one participant received neither technique as they did not have an operation; " is a little confusing, possibly "they" could be exchanged for "the patient"? And, again, a comparison of the results would be informative, were there differences between the groups? Concerning "Table 3. Incidence of significant (>3 or >=4)" or severe pain(>=7) from Visual Analogue Scale Scores": The explanation for superscript 1 is confusing. Do the authors actually mean that "1 RR<0 indicate less incidence with PVB"? Or should it rather be "RR<1 indicates less incidence...". Concerning "Table 4. Results of patient reported outcomes". Again, I get confused by the interpretation of RR values. Do the authors actually mean RR <0, RR>0 and RR<0 for the superscripts 1,2 and 3, respectively? Or should it rather be 1, instead of 0? This table is packed with information and hard to read, possibly the authors could use colours to indicate their positive findings?
--	---

VERSION 3 – AUTHOR RESPONSE

Reviewer's comments	Response

1	The response nicely describes in more detail how the interviews were done, but not which thematic issues were covered, and how the participants were recruited. Thus, it is still challenging for the reader to understand how the results emerged and consequently, for the reader to benefit from the discussion.	We have added the following descriptions to our manuscript to describe how participants were recruited. ‘Patient who were recruited to the trial were asked if they would be willing to be interviewed about their experiences of being in the trial. Those who consented to be interviewed were contacted by the qualitative researcher to arrange a telephone interview. The name and contact details of staff who had been involved in the trail and were willing to be interviewed were provided to the qualitative researcher and an interview was arranged.’ ‘The main themes for patient interviews included the acceptability of the trial; motivations for being involved in the trial; experiences of participating in the trial – for example, did patients feel well-informed about what involvement would entail; thoughts about the questionnaires used to assess levels of pain. The main themes from the staff interviews included experiences of patient identification and screening; experiences of the randomisation process; and reflections on the trial in general terms –what processes went well and were there any areas for improvement.’
2	Given by this difference, I had to choose one of the abstracts to be able to revise the manuscript. I have based my revision on the abstract included in the revised manuscript, not the one presented online.	We can confirm that the abstract online has also been updated correctly according to reviewer’s comments.
3	The paragraph “Participants and follow-up” and the included “Table 1. Baseline characteristics of patients” lack a statistical comparison of the two included study groups - were the groups comparable? This should be included in the text and/ or by a p-value in the table.	We politely disagree that it is appropriate to present results of statistical tests of baseline characteristics in a randomised trial. Because of the use of randomisation any imbalances could only have occurred by chance, so tests would not be appropriate to test ‘comparability’ <Ref: Altman DG, Dore CJ. Randomisation and baseline comparisons in clinical trials. Lancet. 1990;335:149–153>. If it is editorial policy to produce results of such tests we will

		be happy to comply but our preference would be not to include them.
4	Concerning “Table 2. Anaesthetic technique summary”: The explanation for superscript 1, “1 one participant received neither technique as they did not have an operation; “ is a little confusing, possibly “they” could be exchanged for “the patient”? And, again, a comparison of the results would be informative, were there differences between the groups?	In keeping with the style of the manuscript, we have changed the sentence to one participant received neither technique as the participant did not have an operation. With respect to testing for differences between groups in terms of the anaesthetic technique performed - we had no prior hypothesis that the techniques would have the same characteristics (differences were expected), so we do not think a comparison would be necessary or appropriate. We have provided the summary characteristics so the reader can understand how the techniques have been carried out.
5	Concerning “Table 3. Incidence of significant (>3 or >=4)” or severe pain (>=7) from Visual Analogue Scale Scores”: The explanation for superscript 1 is confusing. Do the authors actually mean that “1 RR<0 indicate less incidence with PVB”? Or should it rather be “RR<1 indicates less incidence...”.	We appreciate the reviewer finding this error in the text; it should indeed be RR<1. We have made the change to the footnote to reflect this.
6	Concerning “Table 4. Results of patient reported outcomes”. Again, I get confused by the interpretation of RR values. Do the authors actually mean RR <0, RR>0 and RR<0 for the superscripts 1,2 and 3, respectively? Or should it rather be 1, instead of 0? This table is packed with information and hard to read, possibly the authors could use colours to indicate their positive findings?	These results report differences between group means so the footnote (<0) is correct. However, we agree this was not entirely clear what the comparison was so have added ‘means’ to the column heading.